# Advances in the Bioactivities of Phytochemical Saponins in the Prevention and Treatment of Atherosclerosis

**DOI:** 10.3390/nu14234998

**Published:** 2022-11-24

**Authors:** Huiyu Luo, Junbin Chen, Chuhong Su, Longying Zha

**Affiliations:** Department of Nutrition and Food Hygiene, School of Public Health, National Medical Products Administration (NMPA) Key Laboratory for Safety Evaluation of Cosmetics, Guangdong Provincial Key Laboratory of Tropical Disease Research, Southern Medical University, Guangzhou 510515, China

**Keywords:** atherosclerosis, phytochemicals, saponin, inflammation, oxidative stress

## Abstract

Atherosclerosis (AS) is a chronic inflammatory disease characterized by hardening and narrowing of arteries. AS leads to a number of arteriosclerotic vascular diseases including cardiovascular diseases, cerebrovascular disease and peripheral artery disease, which pose a big threat to human health. Phytochemicals are a variety of intermediate or terminal low molecular weight secondary metabolites produced during plant energy metabolism. Phytochemicals from plant foods (vegetables, fruits, whole grains) and traditional herb plants have been shown to exhibit multiple bioactivities which are beneficial for prevention and treatment against AS. Many types of phytochemicals including polyphenols, saponins, carotenoids, terpenoids, organic sulfur compounds, phytoestrogens, phytic acids and plant sterols have already been identified, among which saponins are a family of glycosidic compounds consisting of a hydrophobic aglycone (sapogenin) linked to hydrophilic sugar moieties. In recent years, studies have shown that saponins exhibit a number of biological activities such as anti-inflammation, anti-oxidation, cholesterol-lowering, immunomodulation, anti-platelet aggregation, etc., which are helpful in the prevention and treatment of AS. This review aims to summarize the recent advances in the anti-atherosclerotic bioactivities of saponins such as ginsenoside, soyasaponin, astra-galoside, glycyrrhizin, gypenoside, dioscin, saikosaponin, etc.

## 1. Introduction

Atherosclerosis (AS) is a long-term chronic and progressive inflammatory disease characterized by hardening and narrowing of arteries. The pathological process of AS is complex and includes at least lipid oxidation and deposition, endothelial injury, inflammation and atherosclerotic plaque formation [1]. AS can lead to a number of arteriosclerotic vascular diseases including cardiovascular diseases (CVDs), cerebrovascular disease (CBVD) and peripheral artery disease (PAD). Around 20 million people worldwide die from atherosclerotic disease each year. CVDs is now the leading cause of global death and mortality is estimated to increase to about 23.6 million by 2030 [2].

AS is a slow, progressive disease that may begin in childhood. AS is more common in middle-aged and elderly people over 40 years old, and progresses faster after 49 years old. The incidence of AS in men is higher than that in women before menopause and the incidence is similar in both sexes after menopause [3,4]. The exact cause of AS is currently not known. Studies have shown that it is a multi-etiological disease, in which multiple factors act at different stages. These factors are known as risk factors which include dyslipidemia, hypertension, diabetes, genetic factors and other factors (physical inactivity, smoking, unhealthy eating habits, etc.) [5]. Early intervention can delay and stop the progression of the disease, and even mildly reverse the atherosclerotic plaque [6]. Changing lifestyle is the basis for the early prevention of AS. Limited total dietary calories, low salt and low fat diet, decreased intake of saturated fatty acids, cholesterol and sugars, and increased intake of soluble dietary fiber are helpful to prevent AS or delay its progression [7]. In recent years, phytochemicals from plant foods (vegetables, fruits, whole grains) and traditional herb plants have been shown to have multiple bioactivities which help to exhibit potential for prevention and treatment against AS [8].

Phytochemicals are a variety of intermediate or terminal low molecular weight secondary metabolites produced during plant energy metabolism [9]. Many types of phytochemicals including polyphenols, saponins, carotenoids, terpenoids, organic sulfur compounds, phytoestrogens, phytic acids and plant sterols have already been identified [10,11,12,13,14]. Saponins are a family of glycosidic compounds consisting of a hydrophobic aglycone (sapogenin) linked to hydrophilic sugar moieties [15]. These naturally-occurred phytochemicals have diversified chemical structures, which is attribute to modifiability in aglycone structure, the attached side chains and the linkage positions of sugar moieties to the aglycone. Saponins can usually be classified into two categories based on the structure of sapogenin: triterpenoid saponins and steroidal saponins (see reviews elsewhere for their chemical structures) [16,17]. Saponins are widely distributed in plants and can be isolated from different parts (seeds, fruits, roots, rhizomes, stems, bark and leaves). Triterpenoid saponins are usually found in legumes, ginseng roots, sunflower seeds, horse chestnut, liquorice roots, spinach leaves, tea leaves, quillaja bark, quinoa seeds, sugar beet or alliums species. Steroidal saponins can often be isolated in oats, yucca, tomato seeds, yam, fenugreek seeds, ginseng roots, asparagus, aubergine and capsicum peppers [16]. In addition, saponins have also been found in some marine animals, such as starfish and sea cucumbers [18]. Accumulated studies have provided evidence to support that saponins exhibit a number of biological activities such as anti-inflammation, anti-oxidation, cholesterol-lowering, immunomodulation, anti-platelet aggregation, etc., which contribute to the prevention and treatment of AS. This review aims to summarize the recent advances in the anti-atherosclerotic bioactivities of saponins in order to better understand the potentials of phytochemical saponins in the prevention and treatment of AS, guiding the next steps in the study and development of agents or functional foods for AS therapy or prevention. Due to the limited length of the paper, saponins (such as ginsenoside, soyasaponin, astra-galoside, glycyrrhizin, gypenoside, dioscin, saikosaponin, etc.) with a relatively sufficient amount of available published studies have been specially chosen for discussion.

## 2. Ginsenosides

Ginsenosides are the main active components of ginseng, which has a long history of clinical application in emergency treatment, CVDs, diabetes, liver and stomach diseases [17] and has also been conventionally used as a functional food and health supplement in China, Korea, Japan and many other countries [19]. More than 50 kinds of ginsenosides have been isolated and identified to date. They are mainly categorized into two forms: pentacyclic triterpenoids (five-ring oleanolic type) and tetracyclic triterpenoids (four-ring dammarane type). The oleanolic type saponins (e.g., Ro) are very low in concentration. The dammarane type saponins can be further divided into propanaxa-diol-type saponins (e.g., Ra1, Ra2, Ra3, Rb1, Rb2, Rb3, Rc, Rd, Rg3, Rh2, F2 and aglycone PD) and propanaxa-triol-type saponins (e.g., Re, Rg1, Rg2, Rh1, Rf, F1 and aglycone PT) [19,20].

A number of studies have reported that ginsenosides (especially the members of the propanaxa-diol group of saponins) exhibit bioactivities in the prevention and treatment of AS. It is quite clear that dyslipidemia, oxidation, inflammation and endoplasmic reticulum (ER) stress play important roles in the pathogenesis of AS. Ginsenosides from Shengmai Yin (SMY), the traditional Chinese herb preparation widely applied in CVD treatments, exerted therapeutic effects on AS by maintaining lipid homeostasis including cholesterol and lysophosphatidyl-choline in Apo E gene knock out (ApoE^−/−^) mice [21]. The ginsenoside-enriched fraction (AP-SF) from Korean ginseng inhibited activation of c-Jun-dependent inflammatory events, indicating its useful role in the prevention of AS, which is a chronic inflammatory disease [22]. Oxidative low-density lipoprotein (ox-LDL) induces endothelium senescence and promotes AS. Panax quinquefolium saponins, isolated from the stems and leaves of the North American form of ginseng (Panax quinquefolium), improved the lipid profile of hyperlipidemic rats and had antioxidant properties, reducing LDL oxidation in cultured rat cardiac myocytes [23]. Ginsenoside Rb1 could alleviate ox-LDL-induced vascular endothelium senescence via the silent information regulator 1 (SIRT1)-Beclin-1-autophagy axis [24]. Ginsenoside Rb1 attenuated tumor necrosis factor α (TNF-α)-induced oxidative stress and inflammatory injury in human umbilical vein endothelial cells (HUVECs) via inhibiting nuclear factor kappa B (NF-κB), c-Jun *N*-terminal kinase (JNK) and p38 signaling pathways [25]. Endothelial dysfunction and senescence play important roles in AS. Ginsenoside Rb1 reduced the hydrogen peroxide (H_2_O_2_)-induced dysfunction of HUVECs by stimulating the sirtuin1/adenosine monophosphate activated protein kinase (AMPK) pathway [26]. Ginsenoside Rb1 effectively blocked the dysfunction of endothelium-dependent vasorelaxation as well as superoxide anion generation and eNOS downregulation induced by homocysteine (an independent risk factor for AS), suggesting that ginseng compounds may have potential clinical applications in controlling homocysteine-associated vascular diseases and other vascular lesions [27]. Ginsenoside Rb1 also exerted anti-angiogenic effects through the peroxisome proliferator activated receptor (PPAR)-gamma (γ) signaling pathway via modulating micro RNA (miR)-33a and pigment epithelial-derived factor (PEDF) expressions [28]. Ginsenoside Rb1 decreased lipid accumulation in macrophage foam cells and increased the stability of atherosclerotic plaque by inducing autophagy in macrophage [29]. Furthermore, ginsenoside Rb1 promoted atherosclerotic plaque stability via facilitating anti-inflammatory M2 macrophage polarization, which is achieved partly by elevating the production of interleukin (IL)-4 and/or IL-13 and the phosphorylation of signal transducer and activator of transcription 6 (STAT6) [30]. Ginsenoside Rb1 attenuated plaque growth and enhanced plaque stability partially by inhibiting adventitial vasa vasorum proliferation and inflammation in ApoE^−/−^ mice [31]. Ginsenoside Rb1 inhibited vascular calcification as a selective androgen receptor modulator as evidenced by exhibiting inhibitory effects on vascular smooth muscle cells (VSMCs) calcification through androgen receptor-mediated Gas6 transactivation and antagonistic effects in prostate cancer cells [32]. More recently, the platelet membrane-cloaked nano-system containing ginsenoside Rb1 were not only capable of improving inflammation such as adhesion inhibition and anti-angiogenesis therapeutic activity in vitro, but also enhancing localization to atherosclerotic plaques in ApoE^−/−^ mice [33]. Ginsenoside Rb2 could ameliorate lipopolysaccharide (LPS)-induced inflammation and ER stress in HUVECs and THP-1 monocytes via the AMPK-mediated pathway [34]. Another study also found that ginsenoside Rb2 produced an anti-inflammatory effect on the process of endothelial cell senescence, suggesting a potential therapeutic effect for AS by targeting miR216a [35]. Ginsenoside Rg3 could alleviate ox-LDL-induced endothelial dysfunction and prevent AS in ApoE^−/−^ mice by regulating PPAR-γ/FAK signaling pathway [36]. Diabetic patients often have low stability of AS because of the unfavorable macrophage polarization and increased inflammation induced by hyperglycemia. Ginsenoside Rg3 could mitigate AS progression in diabetic ApoE^−/−^ mice by switching macrophages to the M2 phenotype, indicating a potential role of Rg3 in preventing and treating diabetic AS [37]. Furthermore, ginsenoside Rg3 stereoisomers differentially inhibited vascular smooth muscle cell proliferation and migration in diabetic AS. This meant that various Rg3 stereoisomers with different chemical structures could lead to significant differential physiological outcomes. The (S)-isomer seemed to have the potential bioactivity to be developed as a promising drug for diabetic AS [38].

Propanaxtriol-type saponins have also been found to exhibit bioactivity for the prevention and treatment of AS. Ginsenoside Rg1 is one of the major components of the formula of FuFang DanShen and XinNaoKang in traditional Chinese medicine. It has been widely used in the prevention and treatment of a broad range of cardiovascular and cerebral-vascular diseases, including AS. It has been shown that XinNaoKang effectively alleviated AS through altering features of the liver metabolite profiles and cecal microbiota [39]. The major components of FuFangDanShen, Ginsenoside Rg1 together with noto-ginsenoside R1 and protocatechuic aldehyde, could decrease AS and ameliorate low-shear stress-induced dysfunction in vascular endothelial cells [40]. Ginsenoside Rg1 itself could significantly suppress apoptosis induced by serum deprivation and, further, effectively induce the autophagic flux by attenuating serum deprivation-induced apoptosis through activating the AMPK/mammalian target of rapamycin (mTOR) signaling pathway in Raw264.7 macrophages [41]. Ginsenoside Rg2 provided direct vascular benefits by inhibiting leukocyte adhesion into the vascular wall and thereby protecting against vascular inflammatory disease such as AS [42]. Ginsenoside Rh1 was able to affect the proliferation, apoptosis and oxidative stress in ox-LDL-treated VECs by activating the nuclear erythroid 2-related factor-2 (Nrf2)/heme oxygenase-1(HO-1) signaling pathway [43]. Ginsenoside F1 could ameliorate inflammatory injury in endothelial cells and prevent AS in mice through A20-mediated inhibition of NF-κB signaling [44].

The natural forms of ginsenosides usually have low absorption efficiency. Nano-formulation has been shown not only to improve the bioavailability of ginsenosides, but also to display excellent biosafety in response to long-term administration [33,45]. Furthermore, ginsenosides need to be metabolized into secondary saponins by the gastrointestinal microbiota before they can be readily absorbed and utilized in the blood. Compound K (CK) is a microbial metabolite of ginsenosides in the intestine. CK exerted anti-atherogenic effects by blunting leukocyte endothelial interaction and transmigration via the negative regulation of NF-κB signaling, because leukocyte endothelial adhesion and transmigration are important steps in the early stage of the pathogenesis of AS [46]. CK was also capable of suppressing the abnormal proliferation and migration of vascular smooth muscle cells, suggesting that CK can be a therapeutic agent in controlling pathologic cardiovascular conditions such as restenosis and AS [47]. CK could attenuate AS formation in mice by activating liver X receptor alpha (LXRα), a potential target for AS therapy [48]. Furthermore, synthesized CK derivatives were able to enhance LXRα activation [49]. CK protected HUVECs against ox-LDL-induced injury through inhibition of NF-κB, p38 and JNK mitogen-activated protein kinase (MAPK) pathways [50]. CK also attenuated ox-LDL-mediated macrophage inflammation and foam cell formation by inducing autophagy and regulating NF-κB, p38 and JNK MAPK signaling [51]. Taken together, these studies suggest that CK is a potential candidate drug for the treatment of AS.

## 3. Soyasaponins

Soyasaponins are a family of phytochemical saponins that are present mainly in soybeans and other legumes (e.g., lentils and green peas) [52]. On average, soybean and its products contain soyasaponins at a content of about 0.17~6.16% [53]. Soyasaponins are amphiphilic oleanane triterpenoid glycosides with sugar moieties (polar side chains) attached to aglycone (a nonpolar pentacyclic ring). According to the chemical structures of aglycones (soya-sapogenol), soyasaponins are now classified into four groups (A, B, E, and DDMP (2, 3-dihydro-2,5-dihydroxy-6-methyl-4H-pyran-4- one)). Group A and B soyasaponins have the biggest abundance of soybean and its products. Group A soyasaponins include members of A1, A2, A3, A4, A5, A6, Aa (acetyl A4), Ab (acetyl A1), Ac, Ad, Ae (acetyl A5), Af (acetyl A2), Ag (acetyl A6), Ah (acetyl A3), and AX. Group B soyasaponins contains members of Ba (V), Bb (I), Bc (II), Bb’(III), Bc’(BX), IV, and Bh soyasaponins [54]. In recent years, accumulated studies have shown that soyasaponins exhibit various health-promoting bioactivities, including anti-atherosclerotic activities.

On the one hand, there is direct evidence supporting the anti-atherosclerotic activities of soyasaponins. Sphytochemical extract containing isoflavones and soyasaponins (SPE) significantly reduced blood glucose, improved atherosclerotic index, and suppressed lipid peroxidation and platelet aggregation in diabetic rats, indicating soyasaponins’ potential usage in prevention and treatment of diabetes mellitus and diabetes-associated AS [55]. By using ApoE^−/−^ mice, a well-recognized animal model for studying AS, soyasaponins (A1 and A2) have been found to decrease the plaque ratio in the aortic root and innominate artery by reducing hypercholesterolemia and inflammation [56]. In addition, total soyasaponins were able to prevent the reduction of blood platelets and fibrinogen and the increase of fibrin degradation products in the intravascular coagulation caused by infusion of endotoxin or thrombin in rats [57].

On the other hand, soyasaponins may exhibit indirect anti-atherosclerotic activities through reducing the risk factors (dyslipidemia, inflammation, oxidation) of AS. First, soyasaponins have long been shown to improve dyslipidemia which is a known important risk factor for AS. Clinical studies have shown that the presence of soyasaponins in food are effective in reducing cholesterol in humans [58,59]. The crude extracts of total soyasaponins could significantly decrease the levels of total cholesterol (TC), low-density lipoprotein cholesterol (LDL-C), and triglyceride (TG), and increase the high-density lipoprotein cholesterol (HDL-C) level, in serum of mice [60]. Group B soyasaponins reduced TC, non-HDL-C, and TG in the plasma of hamsters in comparison with those fed with casein [61]. In high fat diet (HFD)-fed obese mice, soyasaponins (A1 and A2) could reduce TC, TG, and LDL-C, and soyasaponin I could decrease TG and LDL-C. Meanwhile, soyasaponins (A1, A2, and I) all were able to elevate HDL-C [54]. In HFD-fed ApoE^−/−^ mice, soyasaponin A1 reduced TC, TG, and LDL-C, and elevated HDL-C, and soyasaponin A2 lowered TC, TG, and LDL-C in serum [56]. The mechanisms underlying the soyasaponins’ cholesterol-lowering functionalities are associated with their abilities to form insoluble complexes with bile acids and cholesterol and promote its fecal excretion [52,54]. Second, soyasaponins can ameliorate inflammation, which is another key mediator in the pathogenesis of AS. In recent years, the anti-inflammatory activities of soyasaponins have been intensively investigated in different inflammatory models of cells and animals [11,53,54,62] and well reviewed [52,57]. The evidence regarding soyasaponins’ anti-inflammatory activities in macrophages can to a great extent support that soyasaponins may have potentials in the prevention of AS, because macrophages are one of the major players in the pathogenesis of AS. In LPS-stimulated peritoneal macrophages, the crude soyasaponin extracts suppressed the production of prostaglandin E2 (PGE2), nitric oxide (NO), TNF-α and monocyte chemoattractant protein (MCP)-1 in a dose-dependent manner and decreased the mRNA/protein expression levels of cyclooxygenase (COX)-2 and inducible NO synthase (iNOS) by blunting IκB-a degradation [63]. Soyasaponin Ab inhibited NO, PEG2, TNF-a and IL-1β and further weakly inhibited the phosphorylation of ERK, JNK and p38 [62]. In LPS-treated RAW264.7 macrophages, soyasaponins (A1, A2, or I) were able to suppress the release of NO and TNF-a and inhibit the enzymatic activity and mRNA expression of iNOS in a dose-dependent manner via NF-κB inactivation [64]. The underlying mechanisms are related to the inhibition of soyasaponins (A1, A2 or I) on the reactive oxygen species (ROS)-mediated activation of the phosphoinositide 3-kinase (PI3K)/protein kinase B (Akt)/NF-κB signals [65]. Soyasaponin I also suppressed the production of inflammation-related mediator molecules (TNF-a, IL-β, iNOS, NO, COX-2 and PGE2) through inhibiting the IκB-a phosphorylation and the NF-κB nuclear translocation [66]. Moreover, triterpenoid saponins extracted from green vegetable soya beans exerted moderate anti-inflammatory activities by reducing NO release in LPS-stimulated RAW264.7 macrophages [67]. In addition, soyasaponins (A1, A2 or I) could ameliorate metabolic inflammation in HFD-induced obese male C57BL/6 J mice [54]. Third, soyasaponins can reduce oxidative stress, which may be helpful in reducing the risk of AS. The antioxidant ability of soyasaponin has long been recognized, as evidenced by the report that 1 mg/mL of DDMP soyasaponin (bg) had the same extent of abilities to 17.1 units of superoxide dismutase (SOD)/mL on the clearance of superoxide [68]. The scavenging abilities of soyasaponin I on DPPH radicals were found to have a 50% inhibitory concentration (IC_50_) of 70.2 mM, which was comparable to the DPPH radicals clearing activity of a-tocopherol (IC_50_ = 52.1 mM) [66]. Soyasaponins (A1, A2 and I) could not only decrease the LPS-induced ROS generation at a degree equivalent to the classical anti-oxidant *N*-acetyl-L-cysteine (NAC), but also increase SOD activity and the oxidized and reduced glutathione ratio (GSH/GSSG) [65]. In addition, hyper-insulimia might be an important factor in AS. Total soyasaponin (TS) could protect rabbit vascular smooth muscle cells (SMCs) from the high dose of insulin-induced peroxidation and atherogenesis [69].

## 4. Astra-galosides

Astragalus membranaceus belongs to the genus Astragalus of the butterfly flower family [17]. Saponins (astra-galoside), flavonoids and polysaccharides are found to be the major active components of Astragalus membranaceus. Astra-galoside I, II, III, and A (IV) are relatively rich in Astragalus membranaceus. Indirect and direct evidence has shown that astra-galosides exhibit anti-atherosclerotic bioactivities, besides anti-inflammation, anti-oxidation, immunity enhancement, hepatoprotection and antiaging properties [17,70].

First, indirect evidence comes from anti-atherosclerotic studies using astra-galosides-containing Chinese herbs. Danggui buxue decoction (DBD) could affect lipid metabolism in the early stage of AS in diabetic Goto-Kakizaki (GK) rats by significantly reducing homeostasis model assessment of insulin resistance (HOMA-IR), the levels of TG, TC and LDL-C in serum, and the expression of the lipogenic genes monocyte chemotactic protein 1 (MCP-1), Fas, intercellular adhesion molecule 1 (ICAM1) and Cd36 in aorta, as well as significantly increasing the mRNA expression of Scd1 in aorta. Astra-galoside was then identified as one of the absorbed bioactive compounds present in DBD and can be detected in the serum of (GK) rats following administration [71]. Astra-galoside IV was also shown to be the main active components of Buyang Huanwu Decoction (BYHWD) and could inhibit AS by alleviating atherosclerotic inflammation via the inhibition on the activation of Janus tyrosine kinase (JAK)/signal transducer and activator of tranion (STAT) signaling pathway [72]. Astra-galoside IV combined with Tanshinone IIA has been found to significantly decrease lipid areas, increase collagen content and thicken the fibrous cap in the right common carotid arteries in ApoE^−/−^ atherosclerotic mice, and visibly reduce the ox-LDL-induced cytoplasmic lipid droplet accumulation in RAW264.7 macrophages through regulating the PI3K/AKT and TLR4/NF-κB signaling. This indicates that astra-galosides may have the bioactive properties for reinforcing the stability of atherosclerotic plaques [73].

Second, direct evidence comes from anti-atherosclerotic studies using the purified forms of astra-galosides. Purified astra-galoside IV (purity ≥ 90.0%) could alleviate AS through targeting the circ-0000231/miR-135a-5p/chloride intracellular channel 4 (CLIC4) axis in atherosclerotic cell models (ox-LDL-induced HUVECs) [74]. Astra-galoside IV exerted an anti-oxidative stress effect and vascular endothelial protection by decreasing the deposition of lipid in the arterial wall, and these effects could be enhanced when astra-galoside IV was co-treated with salvianolic acid B, the main active ingredients of Salvia miltiorrhiza [75]. Endothelial injury is the main mechanism of AS. Astra-galoside IV could not only prevent ox-LDL-induced injury of endothelial HUVECs by reducing apoptosis, oxidative stress and inflammatory response [76], but also improve the endothelial dysfunction in thoracic aortas from diabetic rats by reducing oxidative stress and calpain-1 [77]. Astra-galoside prevented the occurrence and development of AS by inhibiting vascular inflammation and alleviating endothelial cell injury via the regulation of miR-17-5p and proprotein convertase subtillisin/kexin type 9 (PCSK9)/very low density lipoprotein receptor (VLDLR) signal pathway in VSMCs and ApoE^−/−^ mice [78]. Astra-galoside IV could effectively protect against C/EBP homologous protein-mediated apoptosis by promoting autophagy in ox-LDL-treated macrophages [79]. Astra-galoside IV lowered serum lipid levels, decreased plaque area and increased plaque stability, meanwhile inhibiting inflammation via modulating the MAPK/NF-κB signaling pathway in HFD-induced LDLR^−/−^ mice, indicating that it exhibits anti-atherosclerotic abilities [80].

## 5. Glycyrrhizins

Glycyrrhizin (or glycyrrhizic acid) belongs to the pentacyclic triterpenoid saponins and is a component rich in glycyrrhiza, which is famous as beet root or as a herbaceous plant [17]. The glycyrrhizin-containing Chinese herb XinNaoKang effectively alleviated AS by improving cecal microbiota and lipid metabolism in atherosclerotic mice [31]. Glycyrrhizin is one of the major bioactive components of Ger-Gen-Chyn-Lian-Tang (GGCLT), an officially standardized mixture of Chinese herbal medicines. GGCLT exhibited anti-atherosclerotic action in an ApoE^−/−^ atherosclerotic mice model via the mechanism of activating AMPK and PPARα in hepatocytes, leading to a decrease of lipid formation [81]. Glycyrrhizin has also been identified as one of the bioactive compounds and key chemicals in Yijin-Tang (YJT), which is a traditional prescription for the treatment of hyperlipidemia and AS in traditional Korean medicine [82]. Glycyrrhizin is a known inhibitor of high mobility group box-1 protein (HMGB1) which is a key downstream signal molecule for the activation of NOD-like receptor thermal protein domain associated protein 3 (NLRP3) inflammasome and plays a vital role in VSMC foam cell formation and atherogenesis [83]. In ApoE^−/−^ mice, glycyrrhizin could not only significantly attenuate the LPS/HFD-induced AS and serum levels of HMGB1 [84], but also improve lipid metabolism and suppress vascular inflammation [85].

Enhanced endothelial permeability is a common characteristic of endothelial dysfunction and often found in AS. Elevated levels of trimethylamine-*N*-oxide (TMAO), a novel intestinal microbial metabolite, have been shown to be correlated with endothelial dysfunction and AS. Glycyrrhizin could prevent the TMAO-induced disruption at cell junctions in the endothelial cell monolayers [86].

## 6. Gypenosides

*Gynostemma pentaphyllum* (Thunb.) Makino is an ancient Chinese herbal medicine and tea [87], and is known for its effects in reducing blood pressure, hypolipidemia, hypoglycemia and anti-aging [17]. Gypenoside has been identified as the most important functional component of *Gynostemma pentaphyllum*. There are more than 160 types of gypenoside that have been classified into 12 classes, such as gypenoside III, IV, VII, XII, I, Gyp-A-AH and XLIX, etc. [17,87]. Gypenosides possess a number of bioactivities including anti-atherogenesis, anti-inflammation, anti-cancer, neuroprotection, hepatoprotection and adjusting lipidemia [88,89]. Especially, both gypenoside-containing Chinese herbs or purified gypenosides have been found to exhibit anti-atherosclerotic bioactivity.

The mixture of Hongqu and gypenosides (HG) has been commonly used for centuries in China for the treatment of hyperlipidemia and related diseases including AS. HG is composed of Fermentum Rubrum (Hongqu) and total saponins (gypenosides) of *Gynostemma pentaphyllum* (Thunb.) Makino in a weight ratio of 3.6:1. HG possessed anti-atherosclerotic effects via a multiple action mechanism for regulating blood lipids, anti-inflammation and anti-oxidation in vitamin D3 in a high-fat emulsion-induced atherosclerotic rat model [90,91]. *Gynostemma pentaphyllum* (Thunb.) Makino can decrease the HFD-induced elevation of TMAO, which is considered as a promising new target for AS therapy. Studies using multi-omics approaches revealed that gypenosides reduced plasma TMAO levels by remodeling the microbiota and affecting the trimethylamine lyase needed in the conversion of choline to TMA in intestinal flora, meanwhile interfering with enzymes associated with TCA and lipid metabolism, thus affecting TMAO and lipid metabolism [92]. The anti-atherosclerotic effects of extracted gypenosides were early reported by the study of Qi et al., in which gypenosides exhibited a protective effect on diffuse lipoidosis in liver and AS in hyperlipidemia quails [93]. Increased levels of NO have been shown to play an important role in the pathological process of inflammation and AS. Gypenosides were found to suppress the synthesis of NO in murine macrophages by inhibiting the enzymatic activity of iNOS and attenuating NF-κB-mediated iNOS protein expression [94]. Gypenosides lessened the AS lesion, and reduced the expressions of intercellular cell adhesion molecule-1 (ICAM-1), MCP-1 and NF-κB p65 in aortic wall and serum levels of MDA, ox-LDL, as well as increasing the serum level of total antioxidant capacity, suggesting its anti-atherosclerotic activities via the regulation of inflammation and oxidation [95]. DNA damage contributes to AS while oxidative stress has an important role in the induction of DNA damage. Gypenosides ameliorated cholesterol-induced DNA damage by inhibiting ROS production in HUVECs, suggesting that its inhibitory effect on atherogenesis is correlated with the alleviation of DNA damage [96]. Kruppel-like factor 14 (KLF14) is known as a player in the development of AS. KLF14 exhibits anti-atherogenic effects via the miR-27a-dependent down-regulation of lipoprotein lipase (LPL) and subsequent inhibition of pro-inflammatory cytokine secretion and lipid accumulation. Gypenosides have been identified as the activator of KLF14 and could delay the development of AS in ApoE^−/−^ mice [97]. Furthermore, gypenosides inhibited endothelial cell apoptosis in AS by modulating mitochondria through PI3K/Akt/Bad pathway in ApoE^−/−^ mice [98]. Gypenoside could also inhibit ox-LDL uptake and foam cell formation through enhancing Sirt1-FOXO1 mediated autophagy flux restoration in ox-LDL-treated THP1 cells [99]. Gypenosides alleviated AS through ameliorating endothelial dysfunction via the regulation of the PCSK9/LOX-1 pathway in both VitD3 and high cholesterol diet-induced AS rats and ox-LDL-stimulated HUVECs [100]. More recently, an interesting study compared the bioactivities of gypenosides (Gyp) and heat-processed gypenosides (HGyp) in HFD-fed obese mice and found that they are both valuable for inhibition of obesity, lipid-lowering and metabolic regulation and, especially, the effect of HGyp was better than that of Gyp [101].

Gypenoside XLIX has been identified as a potent activator of PPAR-alpha (α). In human THP-1 monocytic cells, gypenoside XLIX could, in a PPAR-α dependent manner, inhibit the LPS-induced expression and activity of tissue factor which is involved in AS progression [102]. Furthermore, gypenoside XLIX could inhibit the TNF-α-induced expression and activity of vascular cell adhesion molecule-1 (VCAM-1) in HUVECs [103]. More recently, gypenoside XLIX was found to ameliorate HFD-induced AS through regulating intestinal microbiota, alleviating inflammatory response and restraining oxidative stress in ApoE^−/−^ mice [87]. Gypenoside XVII prevented AS by attenuating endothelial apoptosis and oxidative stress via the regulation of the ERalpha-mediated PI3K/Akt pathway in ApoE^−/−^ mice [104].

## 7. Dioscins

Dioscin is a type of spiro-stanol saponin found in Dioscoreaceae, Liliaceae, Caryophyllaceae and Rosaceae plants. Dioscin is composed of diosgenin and a sugar chain on the C-3 hydroxyl group [17]. Recent data suggested that dioscin and its hydrolysis product diosgenin play an anti-atherosclerotic role through their anti-inflammatory, antioxidant, plasma cholesterol-reducing, anti-proliferation and anti-thrombotic effects [105].

Di’ao Xinxuekang (DAXXK) is a pure Chinese medicine herbal preparation refined from dioscin extracted from the roots of Dioscorea panthaica Prain et Burk and Diosorea nipponica Makino. It has been shown that the dioscin-harboring DAXXK exhibits a number of bioactivities (dilating blood vessels, hypotensive, hypolipidemic, improving hemodynamics and anti-platelet aggregation) and is broadly applied in the therapy of various types of CVDs including AS [106]. The chemokines and associated CXCR3 receptor were expressed during inflammation of AS resulting in lymphocytes recruitment and tissue damage. Blockers of the ligand/CXCR3 receptor interaction have potential to develop as anti-atherosclerotic agents. Diosgenin glycosides dioscin showed an IC50 value of 2.1 µM on blocking the CXCR3 receptor interaction of the IP-10 ligand [107]. In HUVECs, dioscin exhibited potent anti-inflammatory effects by suppressing the TNF-α-induced expressions of VCAM-1, ICAM-1 and endothelial lipase (EL) via the NF-κB signaling pathway [108]. In ox-LDL-treated macrophages, dioscin directly decreased foam cell formation, reduced intracellular cholesterol accumulation and lowered the secretion of TNF-a, IL-1β and IL-6 [109]. In HFD-induced AS rat model, dioscin not only lowered the lipids, TNF-a, IL-1β and IL-6 in plasma, but also decelerated atherosclerotic development through reducing atherosclerotic lesion size and aortic lipid level [109]. In potassium oxonate-treated ApoE^−/−^ mice, dioscin could ameliorate hyperuricemia-induced AS by regulating cholesterol metabolism via the farnesoid X receptor (FXR)-signaling pathway [110]. The proliferation and migration of VSMCs are crucial events in the pathological processes of AS and restenosis after percutaneous coronary intervention (PCI). Dioscin was found to attenuate neointima formation in response to balloon injury by inhibiting VSMCs proliferation and migration through the MAPK-FoxM1 pathway, suggesting that dioscin might have potential in the therapy for AS and restenosis after PCI [111]. Disocin exhibited estrogenic activity [112] and was also shown to prevent postmenopausal AS in ovariectomized LDLR^−/−^ mice by inhibiting oxidative stress, inflammation and apoptosis via a mechanism regulating the PGC-1alpha/ERalpha pathway [113].

## 8. Saiko-saponins

Bupleurum (Chai Hu) plants are widely found worldwide and have approximately 200 species [17]. Saiko-saponins are found to be the most important bioactive component of Bupleurum and have been identified as a pentacyclic triterpenoid oleanolic derivative. Saiko-saponins are found only in Bupleurum plants. Several saiko-saponins (a, b, d, c and f) have been separated and identified from Chai Hu plants [114]. Studies have shown that saiko-saponins possess a series of bioactivities, including antiepileptic, antidepressant, anticancer, anti-inflammatory, immunomodulatory, and neuropathic pain relieving activities [115,116].

The anti-atherosclerotic functionalities of saiko-saponins have been witnessed by some studies. Shengxian decoction (SXT) is a well-known traditional Chinese herb formula and has long been clinically considered as an effective formula against CVDs. Saiko-saponins were one of the main active compounds in SXT and shown to contribute to SXT’s therapeutic effects on doxorubicin-induced chronic heart failure in rats [117]. In an in vitro study, saiko-saponins could inhibit the ox-LDL-induced injury of HUVECs by suppressing the ERK1/2 and p38 MAPK signal transduction and possibly via blunting the JNK and p38 MAPK signal transduction [118]. The antiatherogenic actions and possible molecular mechanisms of saikosaponin a was also investigated in the ox-LDL-stimulated THP-1 cells. It was found that saikosaponin a could attenuate ox-LDL uptake and prompt cholesterol efflux through the modulation of the PI3K/Akt/NF-κB/NLRP3 pathway [119]. Saikosaponin a could significantly inhibit adenosine diphosphate (ADP)-induced platelet aggregation and platelet thromboxane formation from arachidonic acid, indicating its therapeutic benefits in the maintenance of vascular homeostasis [120].

XueFuZhuYuTang is a famous traditional herb formula for treating CVDs and related ailments, used in China for centuries. Saikosaponin b and 29-O-acetylsaikosaponin b were found to be components in its decoction by using a high performance liquid chromatography (HPLC)-diode array detection (DAD)-electrospary ionization (ESI)-mass spectrometer (MS) analytical method [121]. Saikosaponin b2 in the concentrations of 5, 15, 25, and 40 μM could remarkably uptake di-octadecylindo-carbocyanine labeled- (DiI) -HDL in HepG2 cells, indicating that it may have potential as a new agent for AS therapy [122].

## 9. Other Saponins

There are studies regarding the anti-atherosclerotic bioactivities of other saponins from plant foods and herbs. Platycodin D, a triterpene saponin from the root of Platycodon grandiflorum A., exhibited anti-atherosclerotic activity in HUVECs partly through its abilities to increase NO levels and to reduce ox-LDL-induced cell adhesion molecule expression and the endothelial adhesion to monocytes [123]. Jujuboside A, a triterpenoid saponin extracted from jujube seeds, could improve cardiac function, alleviate myocardial and endothelial injury and ameliorate dyslipidemia and inflammation by inhibiting the activation of PPAR-α pathway in rats with coronary heart disease [124]. Alfalfa (Medicago sativa) is a leguminous plant with high contents of phytoestrogen and saponins. Dietary administration of alfalfa significantly increased HDL and reduced the formation of fatty streaks in the aorta, right and left coronary arteries in hyperlipidemic rabbits [125].

Afrocyclamin A, an oleanane-type triterpene saponin isolated from Androsace umbellate, exhibited anti-atherosclerotic effects via mediating the p38 MAPK signaling pathway in VSMCs [126]. Araloside C, a bioactive triterpenoid saponin separated from the Chinese herb Aralia elata (Miq.), attenuated foam cell formation and lessened AS by modulating macrophage polarization via Sirt1-mediated autophagy in HFD-fed ApoE^−/−^ mice and ox-LDL-exposed RAW264.7 macrophages [127]. Zygophyllum album saponins prevented atherogenic effect induced by deltamethrin via attenuating arterial accumulation of native and ox-LDL in rats [128]. Saponins extracted from Tribulus terrestris L. could reduce the expression of ICAM-1, VCAM-1 and E-selectin in human endothelial cell lines [129]. Saponin BF523, an active component of Ilex hainanensis, had good inhibitory effect on ox-LDL-induced foam cell formation [130]. Noto-ginsenoside Fc (Fc) is a novel saponin isolated from P. noto-ginseng and has been found to resist platelet aggregation and attenuate high glucose-induced vascular endothelial cell injury by increasing PPAR-γ in diabetic Sprague-Dawley (SD) rats [131]. Elatoside C is a natural saponin isolated from Longya Aralia chinensis L. Elatoside C had anti-oxidative bioactivity and could ameliorate ox-LDL-caused HUVECs injury by promoting autophagy via the elevation of FoxO1 expression, suggesting its potential role in the treatment of AS [132]. Clematichinenoside (AR), a triterpene saponin isolated from the root of Clematis chinensis Osbeck, shows anti-inflammation and antioxidation bioactivities. AR could reduce the expressions of VCAM-1 and ICAM-1 by modulating nicotinamide adenine dinucleotide phosphate (NADPH) oxidase-dependent IKK/NF-κB pathways in TNF-α-induced HUVECs and finally inhibiting monocyte-HUVECs adhesion, indicating that it is potentially valuable for treating early-stage AS [133]. Total saponins extracted from Tribulus terrestris have been reported to protect against AS. This significantly suppressed the angiotensin II-induced increase in cells proliferation, intracellular production of H2O2 and phospho-ERK1/2, mRNA expression of c-fos, c-jun and pkc-α, and the H2O2-induced increase in intracellular free Ca^2+^ [134]. Karaya root saponin exerted a cholesterol-lowering effect in rats fed with high-cholesterol diet [135].

## 10. Outlook

Atherosclerotic vascular diseases bring significant threats to human health. The discovery of food or herb-derived phytochemicals for the prevention and treatment of AS, and understanding their underlying mechanisms, is still in progress. The wide chemical diversity of saponins has attracted much interest in investigation of the health-beneficial bioactivities of these phytochemicals in recent years. Currently, the evidence supporting the anti-atherosclerotic bioactivities of saponins is mostly from animal and in vitro cell studies. Animal and cell study-based evidence is located on the bottom of the Pyramid of Evidence-Based Medicine and has relatively low strength. Therefore, the higher evidential strength of clinical studies such as double blind randomized controlled trials (RCT) regarding the anti-atherosclerotic effects of saponins is awaited in the future. Meanwhile, more animal and cell studies are still needed to discover the anti-atherosclerotic effects of novel saponins and to deepen our understanding of the underlying mechanisms of saponins’ anti-atherosclerotic functionalities.

## Data Availability

Not applicable.

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
