# Peer review of "Advances in the Bioactivities of Phytochemical Saponins in the Prevention and Treatment of Atherosclerosis"

_nutrients, 2022, doi:10.3390/nu14234998_

Round 1

Reviewer 1 Report

The authors submitted an interesting review type of paper, which deals with antiatherosclerotic effect of selected saponins. Since atherosclerosis is a serious disease associated with cardiovascular or neurological impairments, the topic of this manuscript is important and timely.

The authors summarise activities of the saponins against dyslipoproteinemia, inflammatory factors and other processes leading to the atherosclerotic progress. In the paper, special attention is given to ginsenoside, 73 soyasaponin, astragaloside, glycyrrhizin, gypenoside, dioscin, saikosaponin, constituents of plant species used in traditional Chinese medicine. The review is well-arranged and comprehensive.

Nevertheless, in Introduction, I recommend clarifying reasons why the authors selected these saponins and skipped for example detailed activity description of aescin, the major active principle from Aesculus hippocastanum (Hippocastanaceae), which is known for angioprotective effects. Additionally, I recommend the authors addition of chemical structures of saponin aglycones, which are included in the manuscript.

Author Response

Referee 1:

The authors submitted an interesting review type of paper, which deals with antiatherosclerotic effect of selected saponins. Since atherosclerosis is a serious disease associated with cardiovascular or neurological impairments, the topic of this manuscript is important and timely.

The authors summarise activities of the saponins against dyslipoproteinemia, inflammatory factors and other processes leading to the atherosclerotic progress. In the paper, special attention is given to ginsenoside, soyasaponin, astragaloside, glycyrrhizin, gypenoside, dioscin, saikosaponin, constituents of plant species used in traditional Chinese medicine. The review is well-arranged and comprehensive.

Our response: We thank you very much for the referee’s positive comments.

Nevertheless, in Introduction, I recommend clarifying reasons why the authors selected these saponins and skipped for example detailed activity description of aescin, the major active principle from Aesculus hippocastanum (Hippocastanaceae), which is known for angioprotective effects. Additionally, I recommend the authors addition of chemical structures of saponin aglycones, which are included in the manuscript.

Our response: We thank a lot for the referee’s comments and suggestions. Firstly, we chose to summarize the anti-atherosclerotic activities of saponins which have relatively adequate amount of researches. In terms of the referee’s recommendation, we revised this point in the introduction part.

Secondly, as for the aescin that recommended by the referee, we searched the references in Pubmed by using “aescin in abstract or title” combined with “atherosclerosis in any field” and unfortunately found no reference fitting for this. Thus, aescin is not added in the manuscript.

Finally, we agree with the referee that it will make the manuscript perfect if supplementing the chemical structures of saponin aglycones. However, we think it is not necessary. The reasons are 1) the main objective of this review is to discuss the anti-atherosclerotic bioactivities of saponins. We did not talk about the detailed chemical structures of these saponins either the structure-bioactivity relationships. Addition of the chemical structures of saponin aglycones in the revised manuscript will not improve much for our understanding of their anti-atherosclerotic bioactivities. 2) The chemical structures of these saponins and their aglycones have been reviewed elsewhere by other published reviews. We pointed it out in the manuscript. 3) Saponins have diversified chemical structures and usually more than one aglycone. Take soyasaponin as an example, it has three type of aglycones (A, B and E). Therefore, addition of the chemical structures of saponin aglycones in the revised manuscript will increase the unnecessary length of the manuscript.

We thank again for the referee’s comments and suggestions.

Reviewer 2 Report

The presented manuscript is subject-specific and potentially publishable. The review provides abundant information of different saponins groups on their bioactivities which are more or less related to AS prevention or treatment.  

A few comments that would help to better evaluate the information presented.

The aim  of the paper  must reveal what is being aimed by presenting an overview of previous studies.

 A summary of literature data is provided, but I missed the deeper authors' in the context of all studies.

 Please use the plural in headings of saponin groups.

Author Response

Referee 2:

The presented manuscript is subject-specific and potentially publishable. The review provides abundant information of different saponins groups on their bioactivities which are more or less related to AS prevention or treatment.

Our response: We thank you very much for the referee’s positive comments.

A few comments that would help to better evaluate the information presented.

The aim of the paper must reveal what is being aimed by presenting an overview of previous studies.

Our response: We thank you very much for the referee’s comments. We already revised the objective in the introduction part to make it clearer.

A summary of literature data is provided, but I missed the deeper authors' in the context of all studies.

Our response: We thank you very much for the referee’s comments. We only provide a comprehensive summarization of the available studies regarding the anti-atherosclerotic bioactivities of saponins in the main context when we were talking about each saponin. We presented our opinions in the outlook part.

Please use the plural in headings of saponin groups.

Our response: We thank you very much for the referee’s comments. We revised it in the revised manuscript.